# OpenReview forum: "Responsible Diffusion Models via Constraining Text Embeddings within Safe Regions"
_ACM.org/TheWebConf/2025/Conference — WWW 2025 Oral_

### Official Review · Reviewer_9Hkg · 2024-11-19

**Novelty:** 6
**Technical Quality:** 6

**Review:**

Overall, the paper is well written and easy to follow and read. The approach is clearly explained and compared with existing approaches. The related work section is extensive and, most importantly, provides clear insights on how the current approach compares to existing approaches, what aspects are adapted, and what are the limitations in the literature that are addressed. The proposed method is compared with a representative set of baselines and sufficiently evaluated.

On the weaknesses side, there is no critical reflection on the limitations of the approach. The method performs well on one dataset, but recently there have been many datasets published with prompts that lead to unsafe generation of images. The authors should better motivate why more recent datasets have not been used for experimentation. Furthermore, the generalisability of the approach should also be discussed.

Minor comments:
- incomplete sentence at lines 92-93

**Questions:**

- How reliable is the responsible generation? Especially for concepts where it is difficult to understand what fair means, or there is more variation (e.g., consider a prompt similar to "religious artefacts"); what would be a fair generation here and how can it be measured?
- What is the coverage of the unsafe concepts from the literature that are used in this paper?
- What is the coverage of sensitive terms? How are sensitive terms defined?

**Reviewer Confidence:**

2: The reviewer is willing to defend the evaluation, but it is likely that the reviewer did not understand parts of the paper

**Scope:**

3: The work is somewhat relevant to the Web and to the track, and is of narrow interest to a sub-community

---

### Official Review · Reviewer_peMT · 2024-11-27

**Novelty:** 5
**Technical Quality:** 4

**Review:**

The paper explores an approach to enhance safety and fairness in text-to-image diffusion models. The authors propose to learn semantic direction vectors that guide the text prompt away from unsafe content by leveraging the differences in noise estimates during the denoising process. Specifically, they identify these semantic direction vectors in the CLIP embedding space by minimizing the loss between the predicted noise and the implicit classifier's noise estimate for targeted concepts. They then use LoRA to train the direction vector. Their experiments show the effectiveness of the method.

The approach is simple and, to my knowledge, novel. Using LoRA looks like a reasonable choice. Experiments are comprehensive. The paper was overall clear and easy to read.

The authors could discuss more the computational complexity (mentioned in the Appendix) and provide some more structured details on how long it'd take. It's also not clear to me how this would work with multiple text encoders (mentioned in the transferability section). Can the authors provide more details on this? I understand that models are changing quickly, but the latest versions of SD models (available at submission time) should be used in the experiments. Can the authors also provide some examples of the failures of their method? In addition, a more detailed description of how this method differs from SDL would be useful: I understand that the latter uses CFG, but it's unclear to me why the two would lead to very different results.

**Questions:**

Asked above.

**Reviewer Confidence:**

2: The reviewer is willing to defend the evaluation, but it is likely that the reviewer did not understand parts of the paper

**Scope:**

4: The work is relevant to the Web and to the track, and is of broad interest to the community

---

### Official Review · Reviewer_kQiQ · 2024-11-30

**Novelty:** 4
**Technical Quality:** 4

**Review:**

**Paper Summary:**

This paper proposed a novel self-discovery method to identify the specific semantic direction vector in the embedding space. In addition, the author employed a Low-Rank Adaptation (LoRA) for semantic direction vector initialization to reduce the impact on the model performance for other semantics.  The effectiveness of the method is demonstrated by extensive experiments across multiple datasets and models.

**Pros:**

1.The research topic is interesting and very significant.

2.The evaluation experiments are relatively comprehensive and the source code is released to demonstrate reproducibility.


**Cons:**

1. The description of the proposed method is unclear.

2. What is the safe region? How does the method determine if it is the safe region in the paper?

3. There are more similarities between this paper and reference [21], can you explain how this paper addresses the weaknesses in [21]?

4. Lack the explanation of relevant terms, which are not defined before they are used.

5. Regarding fair generation, does the proposed method apply to other attributes (concepts) besides gender and race? For example, age.

**Questions:**

Please refer to my question in the above section.

**Reviewer Confidence:**

3: The reviewer is confident but not certain that the evaluation is correct

**Scope:**

3: The work is somewhat relevant to the Web and to the track, and is of narrow interest to a sub-community

---

### Official Review · Reviewer_rYNY · 2024-12-03

**Novelty:** 5
**Technical Quality:** 5

**Review:**

This paper introduces a novel approach to responsibly prevent the generation of inappropriate images in text-to-image (txt2img) diffusion models by constraining text embeddings to "safe regions" within the embedding vector space. Unlike prior methods that relied on filtering specific keywords—which are vulnerable to adversarial attacks—this approach uses a self-discovery mechanism to identify semantic direction vectors in the embedding space, guiding prompts away from generating unwanted content.

**Strengths**
+ The proposed method addresses an important issue of responsibility in text-to-image generation, introducing an interesting solution that operates on prompt embeddings rather than the raw prompts themselves. This design makes it inherently more robust and effective compared to traditional word-filtering approaches, as demonstrated in the results.
+ Unlike many existing methods, the approach does not add significant computational overhead during image generation and does not rely on costly re-training or fine-tuning of the diffusion models. This makes it practical and scalable for real-world use.
+ The authors’ use of LoRAs for direction vector initialization helps ensure that there is low impact on image quality, and image fidelity and semantic similarity is maintained. As shown by the FID and CLIP scores in conducted evaluations.
+ While the focus of the paper is on preventing inappropriate content, the proposed method is versatile and can be adapted to address other responsibilities, such as reducing biases in image generation. This expands its potential impact beyond the specific use case of inappropriate content prevention.

**Limitations**
- This approach relies on linear relationships between concepts in CLIP, but this may not be the case for all concepts in the embedding space, particularly for more abstract or complex ideas, potentially limiting the method’s robustness.
- The paper only demonstrates the effectiveness of this approach using one txt2img model and specifically CLIP embeddings. It would be helpful to see if these findings can be seen in other embedding spaces or models with different architectures too.
- Image generation models are most popularly used by artists for creative use-cases, and it is possible that their desired content lies near the edges of the ‘safe’ embedding space. This approach may over-restrict the model by reducing the variability in output.
- The paper mentions improved performance against adversarial prompts, but perhaps adversarial prompts can be designed specifically to exploit the way embeddings are manipulated in this approach.

**Questions:**

- The experiments focus on a single text-to-image model and specifically on CLIP embeddings. Have you tested this approach on other text-to-image architectures? If not, how well do you think your solution will generalize to different architectures or embedding spaces with varying properties?

- Given that embedding constraints may restrict the diversity of generated outputs, particularly for creative or boundary-case prompts, how do you ensure that the model remains flexible enough (doesn't overly restrict) for legitimate artistic use cases?

**Reviewer Confidence:**

2: The reviewer is willing to defend the evaluation, but it is likely that the reviewer did not understand parts of the paper

**Scope:**

4: The work is relevant to the Web and to the track, and is of broad interest to the community